# Plasma circulating cell-free mitochondrial DNA in depressive disorders

**Johan Fernström** [1,2]*, **Lars Ohlsson**[3], **Marie Asp**[1,2], **Eva Lavant**[3], **Amanda Holck**[1,2], **Cécile Grudet**[1], **Åsa Westrin**[1,4], **Daniel Lindqvist**[1,4]

**1** Department of Clinical Sciences Lund, Psychiatry, Faculty of Medicine, Lund University, Lund, Sweden, **2** Office for Psychiatry and Habilitation, Psychiatric Clinic Lund, Region Skåne, Sweden, **3** Department of Biomedical Science, Malmö University, Health and Society, Malmö, Sweden, **4** Office for Psychiatry and Habilitation, Psychiatry Research Skåne, Region Skåne, Sweden

* johan.fernstrom@med.lu.se

## Abstract

### Background

Plasma circulating cell-free mitochondrial DNA (ccf-mtDNA) is an immunogenic molecule and a novel biomarker of psychiatric disorders. Some previous studies reported increased levels of ccf-mtDNA in unmedicated depression and recent suicide attempters, while other studies found unchanged or decreased ccf-mtDNA levels in depression. Inconsistent findings across studies may be explained by small sample sizes and between-study variations in somatic and psychiatric co-morbidity or medication status.

### Methods

We measured plasma ccf-mtDNA in a cohort of 281 patients with depressive disorders and 49 healthy controls. Ninety-three percent of all patients were treated with one or several psychotropic medications. Thirty-six percent had a personality disorder, 13% bipolar disorder. All analyses involving ccf-mtDNA were *a priori* adjusted for age and sex.

### Results

Mean levels in ccf-mtDNA were significantly different between patients with a current depressive episode (n = 236), remitted depressive episode (n = 45) and healthy controls (n = 49) (f = 8.3, p<0.001). Post-hoc tests revealed that both patients with current (p<0.001) and remitted (p = 0.002) depression had lower ccf-mtDNA compared to controls. Within the depressed group there was a positive correlation between ccf-mtDNA and "inflammatory depression symptoms" (r = 0.15, p = 0.02). We also found that treatment with mood stabilizers lamotrigine, valproic acid or lithium was associated with lower ccf-mtDNA (f = 8.1, p = 0.005).

### Discussion

Decreased plasma ccf-mtDNA in difficult-to-treat depression may be partly explained by concurrent psychotropic medications and co-morbidity. Our findings suggest that ccf-

**Data Availability Statement:** Data cannot be made freely available as they are subject to secrecy in accordance with the Swedish Public Access to

Information and Secrecy Act. Data requests can be sent to registrator@lu.se and will be subject to a review of secrecy.

**Funding:** This study was supported by the Swedish Mental Health Fund (JF), the Bror Gadelius Memorial Foundation (JF), the Ellen and Henrik Sjöbring Memorial Foundation (JF, DL), the Swedish Research Council (DL) (grant number 2020-01428), and state grants (ALF) from the province of Scania, Sweden (DL, ÅW). The remaining authors report no disclosures or specific funding. The funders had no role in study design, data collection and analysis, decision to publish, or preparation of the manuscript.

**Competing interests:** The authors have declared that no competing interests exist.

mtDNA may be differentially regulated in different subtypes of depression, and this hypothesis should be pursued in future studies.

## Introduction

Oxidative stress and apoptosis trigger the release of mitochondrial DNA (mtDNA) from the cell into the systemic circulation [1]. Circulating cell-free mtDNA (ccf-mtDNA), as measured in blood plasma, triggers inflammatory cascades but may also have beneficial antibacterial effects and contribute to cell-to-cell communications [1]. Increased ccf-mtDNA has been reported in various somatic disorders including sepsis, diabetes, and traumatic injury [2–5]. Recent studies show that also psychological stress may trigger ccf-mtDNA release [1, 6, 7], suggesting that this biomarker might be useful in certain psychiatric disorders.

We have previously shown that unmedicated patients with suicidal [8] and non-suicidal [9] major depressive disorder (MDD) have increased levels of plasma ccf-mtDNA, and that increased ccf-mtDNA levels are associated with hypothalamic-pituitary-adrenal axis hyperactivity [8]. While these findings suggest that depression and suicidality may be accompanied by increased amounts of cellular stress, other studies have reported unchanged [10, 11], or decreased [12] ccf-mtDNA in mood disorders compared to healthy controls. As recently reviewed elsewhere [1], there are several factors relating to assay methodology and study design that might explain divergent findings across studies. Moreover, the use of psychotropic medications may influence mitochondrial function and cellular health [13–15]. For instance, preclinical studies have shown that SSRIs, antipsychotics, and mood stabilizers may improve cellular health and have neuroprotective effects [16–21], but the relationship between these medications and cellular stress marker ccf-mtDNA has not yet been investigated in a real-life clinical sample of depression. Moreover, no previous studies that have investigated the relationship between ccf-mtDNA and specific symptom profiles of depression.

The main aim of the current study was to investigate plasma ccf-mtDNA in a large and diagnostically well-characterized clinical sample difficult-to-treat depression and healthy controls. Moreover, we aimed to test the relationship between ccf-mtDNA and specific symptoms of depression, a history of a suicide attempt and medications that may influence ccf-mtDNA.

## Methods and materials

### Ethical approval

All patients included in the study have given written informed consent to participate. The GEN-DS project was approved by the Regional Ethical Board in Lund, Sweden (2011/673).

### Subject recruitment, patient cohort

This study is a part of a more comprehensive cohort named "Genes, Depression and Suicidality" (GEN-DS), seeking to investigate pharmacogenetic aspects among patients who have made suicide attempts and those who have not. Patients who were previously diagnosed with an affective disorder and had an insufficient treatment response were referred to the GEN-DS study. In this study insufficient treatment response was defined as not having achieved remission with previous and ongoing treatments during the current depressive episode. Recruitment procedures have been described in a previous study [22]. Briefly, 281 patients were referred to the project from four psychiatric clinics in southern Sweden between the years of

2012 and 2020. All referrals of patients with clinical depression according to referring specialist or resident in psychiatry were included in the project. Exclusion criteria were body mass index less than 15, pregnancy or current liver disease.

After inclusion, all patients were diagnosed according to DSM-IV by either a specialist in psychiatry or a senior resident in psychiatry under supervision by a specialist in psychiatry.

The diagnostic procedure included a standardized research protocol including Mini International Neuropsychiatric Interview (MINI) 6.0 [23] and the Structured Clinical Interview for DSM-IV Personality Disorders (SCID-II) [24]. Psychiatric symptoms were assessed using the Comprehensive Psychopathological Rating Scale (CPRS) [25]. We extracted the Montgomery-Åsberg Rating Scale (MADRS) from the CPRS [26]. The structured research protocol also included questions on psychiatric symptoms, suicidal and self-harm behaviour, alcohol and substance use, psychiatric and somatic diagnoses and treatments. Remitted depression was defined as referring only to the nine DSM-IV-TR criterion symptom domains for MDD [27].

## Subject recruitment, controls

Forty-nine healthy controls were recruited through advertisements in social media and through newspaper ads. If deemed eligible for inclusion, controls subjects underwent a MINI. Any previous or present psychiatric illness; addiction disorder; treatment with psychotropic drugs or psychotherapy; somatic illness deemed severe or chronic; ongoing infection; present pregnancy, breast-feeding or treatment with drugs influencing the immune system were considered to be an exclusion criterium. Healthy controls received 500 SEK in compensation after the blood draw.

## Measurement of cell-free mtDNA

Plasma was sampled in the morning after a night of fasting and instructions to avoid taking medications and smoking in the morning. Samples were stored at -80C until analyses.

DNA was isolated from thawed plasma samples using the QIAmp DNA Blood Mini Kit (Qiagen, Valencia, CA, USA) according to the manufacturer's instruction for Blood and body-fluid protocol. Before the isolation of DNA, the plasma samples were centrifuged at 10 000 g for 10 min.

The quantitative analysis of cell-free mtDNA was performed using quantitative real time polymerase chain reaction (PCR). The experiment was run once in triplicate reactions. A dilution series consisting of the PCR product was constructed and used to create a standard curve. The different crossing-point values from the unknown samples were compared with the standard curve, and the corresponding number of mitochondrial units was calculated using the following formula:

The amount of DNA (g μl – 1) was divided with the size of the PCR fragment (bp) and the molar mass per base pair (g mol– 1). The product was finally multiplied with Avogadro's constant. The primers (Life Technologies, Paisley, UK) used for PCR amplification of mtDNA were as stated in the table below:

| Gen Primer forward | Primer reverse | Accession nr |
|---|---|---|
| ND2 CACACTCATCACAGCGCTAA | GGATTATGGATGCGGTTGCT | KJ676545 |

The PCR reactions were carried out using SYBR Green Technology (Thermo Fisher Scientific, Waltham, MA, USA). Each 20 μl reaction contained 5 μl of template, 1 μl of each primer (10 μM), 10 μl SYBR MIX (2 Å~, Sensifast, Bioline, London, UK) and 3 μl of nuclease-free water. Each reaction was run in triplicate on a LC480 LightCycler from Roche,

Mannheim, Germany) using the following program: Initial denaturation at 95˚C for 10 min, followed by 45 cycles consisting of 95˚C in 10 s. for melting, 65˚C for 10 s annealing and 72˚C for 11 s extension. The program ended with a melting curve analysis measuring fluorescence continuously from 60 to 97˚C.

## Statistics

The Statistical Package for the Social Sciences for Mac (SPSS version 27, IBM, Armonk, NY, USA) was used for statistical calculations. One-way ANOVA or Student's t-test were used to compare demographic data between groups. Since the ccf-mtDNA levels were skewed this variable was log-transformed. Bivariate correlations were calculated using Pearson's r or Spearman's Rho, as appropriate. All analyses involving ccf-mtDNA were *a priori* adjusted for age and sex using either ANCOVA or partial correlations. Pearson's chi-2 was used to compare proportions between-groups. All tests were two-tailed and the significance level was set to $p < 0.05$.

Ccf-mtDNA, as a danger-associated molecular pattern–DAMP—may trigger of chronic low-grade inflammation. Systemic low-grade inflammation has been implicated in the pathophysiology of depression [28, 29], particularly in those subjects with symptoms of low energy, fatigue and sleep and appetite disturbances [30]. We therefore calculated a composite symptom score of "inflammatory depression" using CPRS items lassitude, fatiguability, reduced appetite and reduced sleep–and investigated the association between this composite score and ccf-mtDNA.

## Results

### Demographic characteristics

Demographic characteristics and ccf-mtDNA levels in current depression, remitted depression and healthy controls are summarized in **Table 1**.

As expected, MADRS scores were significantly higher among currently depressed individuals compared to those with remitted depression. Treatment with mood stabilizers (either lithium, lamotrigine or valproic acid) was more common in remitted depression than current depression.

Diagnostic characteristics for all patients are summarized in Table 2.

Patients with remitted depression did not fullfill the DSM criteria for a depressive episode at the time of the diagnostic evaluation, although comorbidity with other psychiatric disorders was common as shown in **Table 3** below.

**Table 1. Demographic characteristics and ccf-mtDNA in patients and controls.**

|  | Controls (n = 49) | Current depression (n = 236) | Remitted depression (n = 45) | P-value |
|---|---|---|---|---|
| Age Mean ± SD | 36.7 ± 13.1 | 38.2 ± 13.5 | 34.0 ± 10.2 | 0.13 |
| Sex N (%) females | 36 (73.5) | 154 (65.3) | 30 (66.7) | 0.54 |
| BMI Mean ± SD | 23.8 ± 4.0 | 26.3 ± 5.3 | 25.3 ± 4.7 | 0.006 |
| Current smokers, N (%) | 3 (6.1) | 48 (20.5) | 12 (26.7) | 0.03 |
| MADRS score (mean, SD) | na | 22.9 ± 7.9 | 14.6 ± 9.1 | <0.001 |
| Current treatment with mood stabilizer N (%) | na | 49 (20.8) | 16 (35.6) | 0.03 |
| Current treatment with antipsychotic N (%) | na | 45 (19.1) | 9 (20.0) | 0.90 |
| Current treatment with antidepressant N (%) | na | 192 (81.7) | 32 (71.1) | 0.20 |
| Ccf-mtDNA (C/µl plasma) (mean, SD) | 105 578 ± 207 848 | 35 683 ± 59 906 | 26 762 ± 26 688 | p<0.001 |

Data was missing for BMI (n = 14), smoking (n = 2) and MADRS (n = 17).

**Table 2. Detailed diagnostic characteristics including type of affective disorder.**

| Diagnostic group | Number of patients |
|---|---|
| **Current mood disorder** | 236 |
| Depression, single episode | 8 |
| Recurrent depression | 120 |
| Chronic depression | 79 |
| Depression NOS | 3 |
| Dysthymia | 59 |
| Bipolar disorder, depressive episode | 22 |
| **Remitted mood disorder** | 45 |
| Recurrent depression, in remission | 31 |
| Bipolar depresson, in remissiom | 11 |
| No affective disorder | 3 |

Patients could be assigned to more than one diagnosis.

Ccf-mtDNA was not significantly associated with age (p = 0.70), sex (p = 0.34), BMI (p = 0.66) or smoking (p = 0.74). The most common somatic comorbidities among the patients were musculoskeletal disorders (24.9%), gastrointestinal disorders (16.7%), respiratory disorders (8.5%), cardiovascular disorders (7.1%), neurological disorders (12.5%), endocrinological disorders (13.5%). We calculated a "composite somatic co-morbidity score" in which one point was added for each organ system affected. There was no significant correlation between this score and ccf-mtDNA (p = 0.63). Ccf-mtDNA was not significantly correlated with freezer time (p = 0.68).

## Ccf-mtDNA in patients and controls

Mean levels in ccf-mtDNA were significantly different between patients with a current depressive episode, remitted depressive episode and healthy controls (F = 8.3, p<0.001, adjusting for age and sex,). Post-hoc tests revealed that both patients with current (p<0.001) and remitted (p = 0.002) depression had lower ccf-mtDNA compared to controls. There was no significant difference in ccf-mtDNA between current and remitted depression (p = 0.86). Patient samples had been stored longer in the freezer compared to control samples (median 4 vs 2 years, p<0.001, Mann-Whitney U-test). Storage time was, however, not significantly correlated with ccf-mtDNA as shown above and the group differences in ccf-mtDNA (controls vs ongoing depression and controls vs remitted depression) were still significant even after adding storage time as a covariate (F = 9.3, p<0.001).

Ccf-mtDNA was plotted in current depression, remitted depression and controls in **Fig 1**.

**Table 3. Psychiatric comorbidity in patients with current and remitted depression.**

| | Controls (n = 49) | Current depression (n = 236) | Remitted depression (n = 45) | P-value |
|---|---|---|---|---|
| Bipolar disorder N (%) | na | 24 (10.2) | 12 (26.7) | 0.002 |
| Personality disorder N (%) | na | 81 (34.3) | 20 (44.4) | 0.53 |
| Anxiety disorder N (%) | na | 132 (55.9) | 24 (53.3) | 0.49 |
| Substance or alcohol use disorder N (%) | na | 12 (5.1) | 2 (4.4) | 0.82 |

Chi-square tests were carried out to compare current depression vs remitted depression

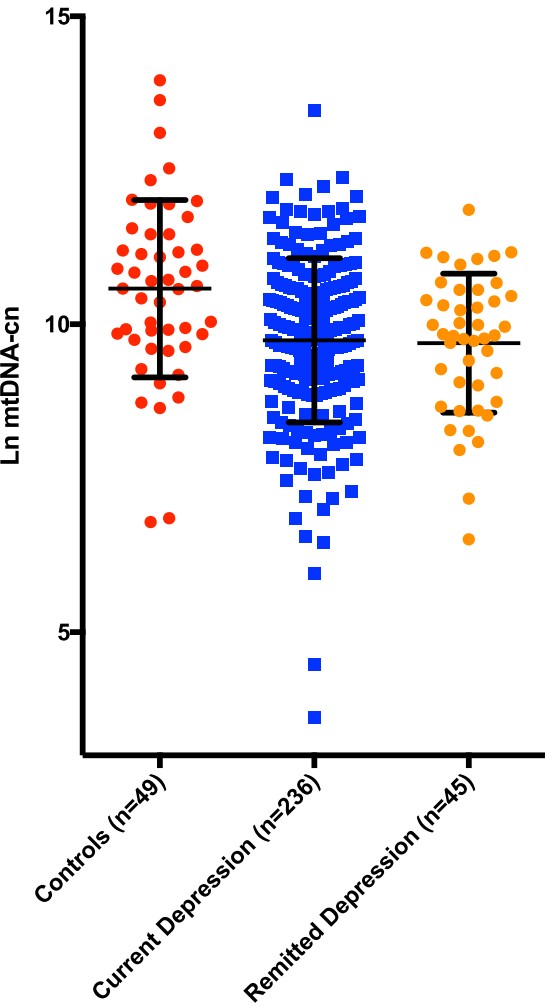

**Fig 1. Log transformed ccf-mtDNA in controls, current depression and remitted depression.** Error bars represent mean, SD. The group effect was significant (F = 8.3, p<0.001, adjusted for age and sex). Post-hoc tests revealed that patients with current (p<0.001) and remitted (p = 0.002) depression had lower ccf-mtDNA compared to controls.

## Associations between ccf-mtDNA and psychotropic medications

Patients taking mood stabilizers (n = 65) had significantly lower ccf-mtDNA compared to those not taking mood stabilizers (n = 216) (F = 8.1, p = 0.005, adjusted for age and sex). There were no differences in ccf-mtDNA between those taking antidepressants or not, or between those taking antipsychotics or not (all p>0.27). Patients with bipolar disorder did not differ significantly in ccf-mtDNA compared to patients without bipolar disorder (p = 0.42).

Nineteen patients (6.8%) took no psychotropic medications at the time of the diagnostic evaluation. There was no significant difference in ccf-mtDNA between those patients and all other patients (p = 0.28).

## Associations between ccf-mtDNA and psychiatric symptoms and suicidality

Ccf-mtDNA was not significantly correlated with MADRS or the SUAS subscale comprising items 16–20 (all p>0.25). As shown in **Fig 2**, the inflammatory depression composite score

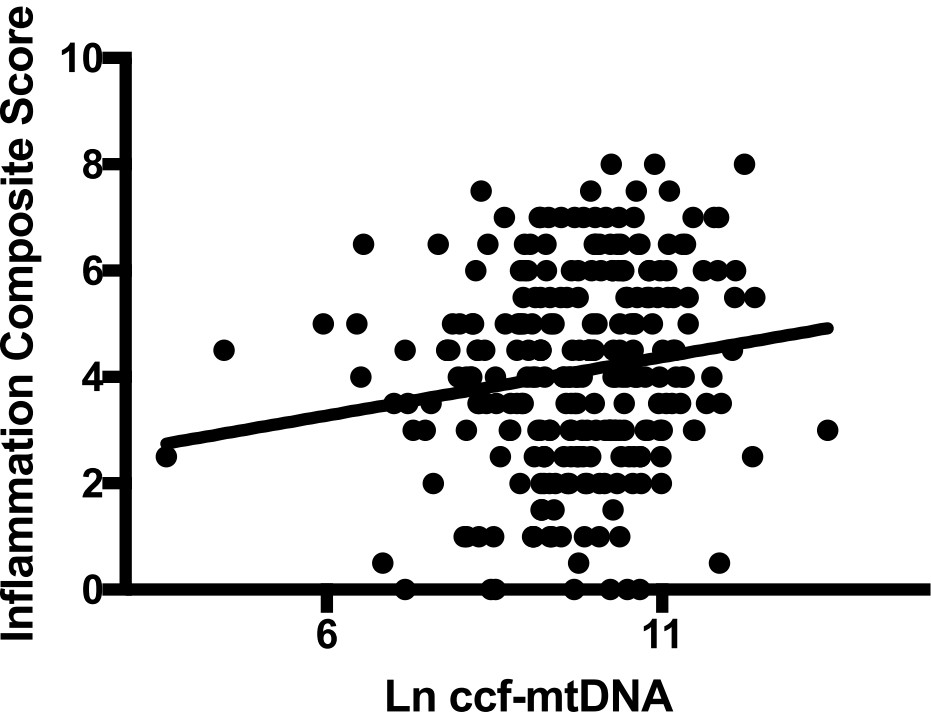

**Fig 2. Correlation between ccf-mtDNA and "inflammation composite score" calculated by summarizing CPRS-items "lassitude", "fatiguability", "reduced appetite" and "reduced sleep".** Both patients with current and remitted depression were included. The correlation was significant (r = 0.15, p = 0.02, df = 264, adjusting for age and sex).

correlated positively and significantly in all patients (r = 0.15, p = 0.02, df = 264, adjusted for age and sex). Patients with a history of a suicide attempt (n = 83) did not differ in ccf-mtDNA levels compared to those who had not made a suicide attempt (n = 191) (p = 0.18, adjusting for age and sex). Information regarding previous suicide attempts was missing for seven patients.

## Discussion

Our most salient finding was that patients with depression in secondary psychiatric care had lower ccf-mtDNA levels compared to healthy controls. These findings contrast our previous reports on unmedicated and suicidal depressed patients [8, 9], but are more in line with other studies on mood disorders [10–12]. Although low ccf-mtDNA was found in the depressed group overall, an "inflammatory depression symptom profile" [30] was conversely associated with higher ccf-mtDNA, suggesting that ccf-mtDNA may be differentially regulated across depression symptom profiles. Finally, treatment with mood stabilizers was associated with the lowest ccf-mtDNA within the depressed group, consistent with animal studies showing that these medications may promote cellular and neuronal health [31].

Previous studies have reported inconsistent findings regarding between-group differences in ccf-mtDNA in patients with mood disorders and healthy controls. The divergent results across studies may be explained by cohort-specific factors such as medication status, illness chronicity, symptom profiles and somatic and psychiatric co-morbidity. In the present study, 93% medicated with one, or a combination of several, psychotropics. Moreover, somatic as well as psychiatric comorbidities were common in the current sample. These sample

characteristics are largely in contrast to one of our previous studies, where we reported *increased* ccf-mtDNA in depression, in which all patients were unmedicated, somatically healthy and had no or minimal psychiatric co-morbidity [9]. In the other study from our group in which increased ccf-mtDNA was reported patients versus controls, all patients were also unmedicated and had recently attempted suicide at the time of the blood sampling [8]. In line with the findings of the present study, Kageyama et al. also reported decreased ccf-mtDNA in unipolar and bipolar depression compared to controls [12]. In another recent study, Jeong et al. found no difference in ccf-mtDNA between adolescents with bipolar disorder and healthy controls [10]. Moreover, Jeong et al. also reported that severity of depressive symptoms was negatively correlated with ccf-mtDNA, although specific depression symptom profiles were not investigated in this partciluar study. In both of these studies [10, 12], a substantial part of the patients medicated with one or more psychotropic. In the present study, we found, in exploratory analyses, that patients within the depressed group treated with mood stabilizers had the lowest levels of ccf-mtDNA, suggesting that at least part of the between-group differences in ccf-mtDNA may be accounted for by medication status. Psychotropic medications, and lithium in particular, are known to influence mitochondrial biology and cellular health [20]. In one study based on pluripotent stem cell technology, mitochondrial abnormalities were found in neurons from patients with bipolar disorders. Furthermore, lithium could normalize some of these alterations [32]. Similar effects of lithium were shown in a post-mortem study, in which activity of electron transport chain (ETC) enzyme complex I-III was increased in human frontal cortex after exposure to lithium [33]. Moreover, an animal model of mania, in which mitochondrial dysfunction was induced using d-amphetamine, both lithium and valproate were found to reverse some of these alterations [34]. Also lamotrigine may have neuroprotective effects mediated via its actions on mitochondrial function, according to some preclinical studies [35, 36]. We show, for the first time, that medication with mood stabilizers is associated with lower plasma levels of ccf-mtDNA in a large clinical sample of depression. These findings may have future implications for treatment response and the understanding of mechanistic actions of mood stabilizers. Such issues should be pursued in future studies.

Both clinical and preclinical studies suggest that ccf-mtDNA is an immunogenic molecule [37–41], although this assumption has also recently been challenged [1]. Mitochondrial DNA can act as damage associated molecular patterns (DAMPs) triggering the innate immune response primarily through binding to the toll-like receptor 9 (TLR-9). Interestingly, we found a significant positive relationship between ccf-mtDNA and depressive symptoms that have been more closely linked to low-grade inflammation [30]. While highly preliminary and in need of replication, these findings suggest that ccf-mtDNA may be differentially regulated in different subtypes of depression. Specifically, our findings point to a role of ccf-mtDNA in "inflammatory depression"; a depression subtype that has been associated with worse treatment response to conventional SSRIs and a better treatment anti-inflammatory compounds [29, 42]. Future clinical trials testing the antidepressant efficacy of such interventions should consider measuring ccf-mtDNA as a potential biomarker of treatment response.

While the present study has several notable strengths including the large sample size and the thorough diagnostic assessments, it also comes with several limitations. There are many factors that potentially may influence ccf-mtDNA levels that were not accounted for in the present study. As described above, both exercise and psychological stress can induce changes in ccf-mtDNA within minutes [6, 7]. We attempted to mitigate these effects by standardizing blood sampling procedures to be done fasting in the morning. We did not, however, record levels of subjective perceived stress preceding the blood draw, sleep patterns or other health behaviors, thus we can not rule out that this may have influenced our results. Sample storage

time in freezer differed significantly between patients and controls, but was not significantly correlated with ccf-mtDNA. Moreover, the main group effects remained significant even after taking this factor into account, making it unlikely that storage time confounded our results. The current study was originally designed to test another primary hypothesis, namely the relationship between genetic variants and suicidal behavior. Thus, we did not, *a priori*, power the study with the main intent to investigate differences in ccf-mtDNA between patients and controls. Although this might be considered a weakness of the study, our previous experiences using the same biomarker assay in other MDD/control cohorts (with substantially smaller sample size than the present study) showed effect sizes ranging between Cohen's d of 0.9 [9] to Cohen's d >2 [8]. Therefore, we believed that the current sample size would be large enough to detect a significant group difference between patients and controls. Finally, while we argue above that psychotropic medication might be a factor leading to lower ccf-mtDNA in the patient group, we did not find a significant difference in ccf-mtDNA between a small subset of the patients not taking any medications (7%) and all other patients. Potential reasons for this include that the unmedicated group might have been too small to detect a significant effect, and future larger studies should further investigate this hypothesis.

In conclusion, we found that plasma ccf-mtDNA is decreased in individuals with depressive disorders compared to controls. These findings may be partly explained by concurrent psychotropic medications and psychiatric co-morbidity. Specifically, our results suggest that treatment with mood stabilizers may affect ccf-mtDNA levels, possibly as a down-stream consequence of their effect on mitochondrial enzymatic processes. Our findings suggest that ccf-mtDNA may be differentially regulated in different subtypes of depression. This hypothesis might be valuable to pursue in future studies.

## Author Contributions

**Conceptualization:** Johan Fernström, Åsa Westrin, Daniel Lindqvist.

**Data curation:** Cécile Grudet, Åsa Westrin.

**Formal analysis:** Johan Fernström, Daniel Lindqvist.

**Funding acquisition:** Johan Fernström, Åsa Westrin, Daniel Lindqvist.

**Investigation:** Johan Fernström, Marie Asp, Amanda Holck, Åsa Westrin.

**Methodology:** Johan Fernström, Lars Ohlsson, Eva Lavant.

**Project administration:** Åsa Westrin, Daniel Lindqvist.

**Resources:** Åsa Westrin.

**Software:** Cécile Grudet.

**Supervision:** Lars Ohlsson, Åsa Westrin, Daniel Lindqvist.

**Visualization:** Johan Fernström, Daniel Lindqvist.

**Writing – original draft:** Johan Fernström, Daniel Lindqvist.

**Writing – review & editing:** Johan Fernström, Marie Asp, Amanda Holck, Cécile Grudet, Åsa Westrin, Daniel Lindqvist.

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
