## [Decision Letter · Decision Letter 0]

21 Jun 2021

PONE-D-21-17236

Plasma circulating cell-free mitochondrial DNA in depressive disorders

PLOS ONE

Dear Dr. Fernström,

Thank you for submitting your manuscript to PLOS ONE. After careful consideration, we feel that it has merit but does not fully meet PLOS ONE’s publication criteria as it currently stands. Therefore, we invite you to submit a revised version of the manuscript that addresses the points raised during the review process.

We look forward to receiving your revised manuscript.

Kind regards,

Tadafumi Kato

Academic Editor

PLOS ONE

Journal Requirements:

3. Please ensure that you refer to Figure 2 in your text as, if accepted, production will need this reference to link the reader to the figure.

Additional Editor Comments (if provided):

Two experts assessed the paper, and both found the importance of the study and suggested several issues to further improve the paper. Please follow the advise of the referees.

Reviewers' comments:

Reviewer's Responses to Questions

**Comments to the Author**

1. Is the manuscript technically sound, and do the data support the conclusions?

Reviewer #1: Yes

Reviewer #2: Yes

2. Has the statistical analysis been performed appropriately and rigorously? 

Reviewer #1: Yes

Reviewer #2: Yes

3. Have the authors made all data underlying the findings in their manuscript fully available?

Reviewer #1: Yes

Reviewer #2: Yes

4. Is the manuscript presented in an intelligible fashion and written in standard English?

Reviewer #1: Yes

Reviewer #2: Yes

5. Review Comments to the Author

Reviewer #1: The authors investigated whether the plasma circulating cell-free mitochondrial DNA (ccf-mtDNA) levels differ between patients with clinical depression and healthy control in a large sample set. They showed an overall decrease in plasma ccf-mtDNA levels in patients compared to controls. This paper is well written. The study is quite interesting and significantly adds to the literature of the field. The following points should be addressed before publication.

Major comments:

1)Table 1:

As for the ref. 22, all the patients of ‘Current depression’ and ‘Remitted depression’ groups were diagnosed with some types of affective disorders.

In table 1, it is hard to understand which affective disorder was comorbid with other psychiatric disorders.

Since the number of patients in this paper increased (n = 279) compared to those of ref. 22 (n = 274), and to make a better understanding, please omit the row from ‘Bipolar disorder’ to ‘Substance or alcohol use disorder’ in table 1 and create new tables similar to table 2 and 3 of ref. 22.

2)-4)

Since the ‘Current depression’ and ‘Remitted depression’ groups include major depressive disorder, bipolar disorder, and other types of affective disorders such as dysthymia and mixed anxiety and depressive disorder. The plasma ccf-mtDNA level might be affected by the ratio of the types of affective disorders.

2) Is there any difference in plasma ccf-mtDNA level among major depressive disorder patients with depressed state, major depressive disorder patients with remitted state, and controls?

3) Is there any difference in plasma ccf-mtDNA level among bipolar disorder patients with depressed state, bipolar disorder patients with remitted state, and controls?

4) Is there any difference in plasma ccf-mtDNA level between major depressive disorder patients taking mood stabilizers and those not taking mood stabilizers?

Minor comment

1) Line 74

Please describe the definition of ‘difficult-to-treat depression’ briefly.

If the definition of ‘difficult-to-treat depression’ is the same as of the ref.22, please cite the manuscript.

2) Line 91:

What does ‘clinical depression’ mean?

3) Line 178-179

Does ‘Patients with remitted 179 depression wadid not fulfill the DSM criteria’ mean

‘Patients with remitted 179 depression did not fulfill the DSM criteria’?

Reviewer #2: This study investigated ccf-mtDNA in medicated patients with depressive disorders and healthy controls. The main finding is that ccf-mtDNA is decreased in medicated patients and that ccf-mtDNA is correlated with "inflammatory depression composite score". I believe that the findings are novel and interesting, the methodology seems sound and the paper is well-written. Here are my comments:

1- Did the author calculate study power? Also, the number of controls are much lower than cases. Is that a problem to draw the conclusions?

2- if there is a difference in the handling time between patients and controls?

3- Would be interesting to examine the correlation between the amounts of medication taken and the level of ccf-mtDNA. Do the authors have this information available?

4- Based on literature, would you expect that the patients that were not under medication (6.8%) to have higher levels of ccf-mtDNA?

5- "Damage associated molecular patterns (DAMPs) are expressed on the surface of mitochondrial DNA"... is that correct?

6. PLOS authors have the option to publish the peer review history of their article (what does this mean?). If published, this will include your full peer review and any attached files.

Reviewer #1: **Yes: **Yuki Kageyama

Reviewer #2: No

---

## [Author Response · Author response to Decision Letter 0]

21 Sep 2021

Comment from editor

Please ensure that you refer to Figure 2 in your text as, if accepted, production will need this reference to link the reader to the figure.

Authors response: We now refer to Figure 2 in the text in the results section.

Comments from Reviewer #1: The authors investigated whether the plasma circulating cell-free mitochondrial DNA (ccf-mtDNA) levels differ between patients with clinical depression and healthy control in a large sample set. They showed an overall decrease in plasma ccf-mtDNA levels in patients compared to controls. This paper is well written. The study is quite interesting and significantly adds to the literature of the field. The following points should be addressed before publication.

Major comments:

1) Table 1: As for the ref. 22, all the patients of ‘Current depression’ and ‘Remitted depression’ groups were diagnosed with some types of affective disorders.

In table 1, it is hard to understand which affective disorder was comorbid with other psychiatric disorders.

Since the number of patients in this paper increased (n = 279) compared to those of ref. 22 (n = 274), and to make a better understanding, please omit the row from ‘Bipolar disorder’ to ‘Substance or alcohol use disorder’ in table 1 and create new tables similar to table 2 and 3 of ref. 22.

Authors response: Thanks very much for this comment. The sample in reference 22 (Asp et al., Plos One) is overlapping (but not identical) to the sample in the present study. In the present study we have included all patients recruited for the study described by Asp et al (“GEN-DS”) for whom there were plasma samples and adequate clinical information available at the time of our analyses of ccf-mtDNA. 

In response to this comment we have now added to additional Tables (2+3). In table 2 we give a more detailed description of which subtypes of affective disorders the patients were diagnosed with. In Table 3, we show psychiatric co-morbidity in the two groups. Since the main purpose of these Tables is to give the reader a sense of the clinical characteristics of the sample, and specifically how the two main groups in or analyses (current vs remitted depression) are balanced with regards to these variables, we believe it makes more sense to show comorbidity numbers in these two, broader, groups rather than in several sets of smaller subgroups that are not a part of the main analyses. 

2)-4)

Since the ‘Current depression’ and ‘Remitted depression’ groups include major depressive disorder, bipolar disorder, and other types of affective disorders such as dysthymia and mixed anxiety and depressive disorder. The plasma ccf-mtDNA level might be affected by the ratio of the types of affective disorders.

2) Is there any difference in plasma ccf-mtDNA level among major depressive disorder patients with depressed state, major depressive disorder patients with remitted state, and controls? 

Authors response: There was no significant difference in ccf-mtDNA between unipolar MDD patients with depressed state and unipolar MDD patients with remitted state (p=0.39, one-way ANOVA), but both groups differed significantly from controls (p<0.001 and p=0.01). ccf-mtDNA was lower in both MDD groups compared to controls, just as in our main analyses. 

3) Is there any difference in plasma ccf-mtDNA level among bipolar disorder patients with depressed state, bipolar disorder patients with remitted state, and controls?

Authors response: There was no significant difference in ccf-mtDNA between bipolar depressed patients with depressed state and bipolar patients with remitted state (p=0.62 one-way ANOVA), but both groups differed significantly from controls (p=0.01). ccf-mtDNA was lower in both bipolar groups compared to controls, just as in our main analyses.

4) Is there any difference in plasma ccf-mtDNA level between major depressive disorder patients taking mood stabilizers and those not taking mood stabilizers?

Authors response: Yes , in line with the findings that we present when all subjects are included, unipolar MDD patients with mood stabilizers had lower ccf mtDNA than those without mood stabilizers (p=0.01).

Minor comment

1) Line 74

Please describe the definition of ‘difficult-to-treat depression’ briefly.

If the definition of ‘difficult-to-treat depression’ is the same as of the ref.22, please cite the manuscript.

Authors response: Thank you very much for this comment. We have now clarified in Methods section on page 5 that:

“Patients who were previously diagnosed with an affective disorder and had an insufficient treatment response were referred to the GEN-DS study. In this study, insufficient treatment response was defined as not having achieved remission with the previous and ongoing treatments during the current depressive episode.” 

Please note that, for some patients, the depressive episodes were in fact judged to be in remission after detailed diagnostic assessments. The definition above refers to the instructions for the clinicians when referring patients to the study. 

2) Line 91:

What does ‘clinical depression’ mean?

Authors response: We have revised this section (see also above) to be more consistent with the description in ref #22 (Asp et al). Please see “subject recruitment, patient cohort” in Methods section.

3) Line 178-179

Does ‘Patients with remitted 179 depression wadid not fulfill the DSM criteria’ mean

‘Patients with remitted 179 depression did not fulfill the DSM criteria’?

Authors response: Yes, thanks for noticing this typo, this has been corrected. 

Comments from Reviewer #2: This study investigated ccf-mtDNA in medicated patients with depressive disorders and healthy controls. The main finding is that ccf-mtDNA is decreased in medicated patients and that ccf-mtDNA is correlated with "inflammatory depression composite score". I believe that the findings are novel and interesting, the methodology seems sound and the paper is well-written. Here are my comments:

1- Did the author calculate study power? Also, the number of controls are much lower than cases. Is that a problem to draw the conclusions?

Authors response: The study was originally designed to test another primary hypothesis (the relationship between genetic variants and suicidal behavior). Thus, we did not, a priori, power the study with the main intent to test the hypotheses of the present study. However, previous experiences using the same biomarker assay in other MDD/control cohorts (with substantially smaller sample size than the present study) showed effect sizes ranging between Cohen’s d of 0.9 (Lindqvist et al., 2018, Neuropsychopharmacology) to Cohen’s d >2 (Lindqvist et al., 2016, Translational Psychiatry). Therefore, we deemed that the current patient and control groups would be large enough to detect a significant group difference between patients and controls. As suggested by the reviewer, similar group sizes in patients vs controls would yield greater power. In our analysis, the 95% CI for mean (log-transformed ccf-mtDNA) was 10.2-11.0 (controls), 9.6-9.9 (current depression) and 9.4-10.0 (remitted depression).

2- if there is a difference in the handling time between patients and controls?

Authors response: Thanks for this comment. Patient samples had been stored longer in the freezer compared to control samples (median 4 years vs median 2 years, p<0.001, Mann-Whitney U test). Storage time was, however, not significantly correlated with ccf-mtDNA (p=0.68) and the group differences in ccf-mtDNA (controls vs ongoing depression and controls vs remitted depression) were still significant even after adjusting for storage time (f=9.3, p<0.001). Thus, we do not believe that differences in storage time had a significant impact on our results. This information has been added to the Results section.

3- Would be interesting to examine the correlation between the amounts of medication taken and the level of ccf-mtDNA. Do the authors have this information available?

Authors response: Yes – that would indeed be interesting. However, we do not, at this time, have the required data regarding medication dosage or blood concentration. Analysis of how the number of different psychotropic medications correlate to ccf-mtDNA yielded no significant results. 

4- Based on literature, would you expect that the patients that were not under medication (6.8%) to have higher levels of ccf-mtDNA?

Authors response: Yes, based on our previous studies that would be expected. However, this has never been tested before. In our sample, it was only a very small subset of patients without medications, hence we might not have been able to detect such an effect. We have added a sentence about this in the limitations section (page 16):

“Finally, while we argue above that psychotropic medication might be a factor leading to lower ccf-mtDNA in the patient group, we did not find a significant difference in ccf-mtDNA between a small subset of the patients not taking any medications (7%) and all other patients. Potential reasons for this include that the unmedicated group might have been too small to detect a significant effect, and future larger studies should further investigate this hypothesis. “

5- "Damage associated molecular patterns (DAMPs) are expressed on the surface of mitochondrial DNA"... is that correct?

Authors response: Thanks for noticing this. We have now revised:

“Mitochondrial DNA can act as damage associated molecular patterns (DAMPs) triggering the innate immune response primarily through binding to the toll-like receptor 9 (TLR-9)”

---

## [Decision Letter · Decision Letter 1]

19 Oct 2021

PONE-D-21-17236R1Plasma circulating cell-free mitochondrial DNA in depressive disordersPLOS ONE

Dear Dr. Fernström,

Thank you for submitting your manuscript to PLOS ONE. After careful consideration, we feel that it has merit but does not fully meet PLOS ONE’s publication criteria as it currently stands. Therefore, we invite you to submit a revised version of the manuscript that addresses the points raised during the review process.

We look forward to receiving your revised manuscript.

Kind regards,

Tadafumi Kato

Academic Editor

PLOS ONE

Journal Requirements:

Additional Editor Comments (if provided):

The essence of the other discussions made based on the comments by the referee 2 should also be incorporated into the manuscript, i.e., statistical power (point 1), handling time (point 2).

Reviewers' comments:

Reviewer's Responses to Questions

**Comments to the Author**

1. If the authors have adequately addressed your comments raised in a previous round of review and you feel that this manuscript is now acceptable for publication, you may indicate that here to bypass the “Comments to the Author” section, enter your conflict of interest statement in the “Confidential to Editor” section, and submit your "Accept" recommendation.

Reviewer #1: All comments have been addressed

2. Is the manuscript technically sound, and do the data support the conclusions?

Reviewer #1: Yes

3. Has the statistical analysis been performed appropriately and rigorously? 

Reviewer #1: Yes

4. Have the authors made all data underlying the findings in their manuscript fully available?

Reviewer #1: Yes

5. Is the manuscript presented in an intelligible fashion and written in standard English?

Reviewer #1: Yes

6. Review Comments to the Author

Reviewer #1: (No Response)

7. PLOS authors have the option to publish the peer review history of their article (what does this mean?). If published, this will include your full peer review and any attached files.

Reviewer #1: **Yes: **Yuki Kageyama

---

## [Author Response · Author response to Decision Letter 1]

20 Oct 2021

Comment from editor:

The essence of the other discussions made based on the comments by the referee 2 should also be incorporated into the manuscript, i.e., statistical power (point 1), handling time (point 2).

Authors response: Thanks for this comment. We have now added, to the discussion section (p17):

“The current study was originally designed to test another primary hypothesis, namely the relationship between genetic variants and suicidal behavior. Thus, we did not, a priori, power the study with the main intent to investigate differences in ccf-mtDNA between patients and controls. Although this might be considered a weakness of the study, our previous experiences using the same biomarker assay in other MDD/control cohorts (with substantially smaller sample size than the present study) showed effect sizes ranging between Cohen’s d of 0.9 (9) to Cohen’s d >2 (8). Therefore, we believed that the current sample size would be large enough to detect a significant group difference between patients and controls.”

In the results section (page 12) we report associations between ccf-mtDNA and handling time and we have now also added one sentence about this in the Discussion (page 16-17):

“Sample storage time in freezer differed significantly between patients and controls, but was not significantly correlated with ccf-mtDNA. Moreover, the main group effects remained significant even after taking this factor into account, making it unlikely that storage time confounded our results.”

---

## [Editor Report · Decision Letter 2]

22 Oct 2021

Plasma circulating cell-free mitochondrial DNA in depressive disorders

PONE-D-21-17236R2

Dear Dr. Fernström,

We’re pleased to inform you that your manuscript has been judged scientifically suitable for publication and will be formally accepted for publication once it meets all outstanding technical requirements.

Kind regards,

Tadafumi Kato

Academic Editor

PLOS ONE
---

## [Editor Report · Acceptance letter]

27 Oct 2021

PONE-D-21-17236R2 

Plasma circulating cell-free mitochondrial DNA in depressive disorders 

Dear Dr. Fernström:

I'm pleased to inform you that your manuscript has been deemed suitable for publication in PLOS ONE. Congratulations! Your manuscript is now with our production department. 

Kind regards, 

on behalf of

Dr. Tadafumi Kato 

Academic Editor

PLOS ONE